# Combined Lifestyle Interventions in the Prevention and Management of Asthma and COPD: A Systematic Review

**DOI:** 10.3390/nu16101515

**Published:** 2024-05-17

**Authors:** Charlotte D. C. Born, Rohini Bhadra, George D’Souza, Stef P. J. Kremers, Sucharita Sambashivaiah, Annemie M. W. J. Schols, Rik Crutzen, Rosanne J. H. C. G. Beijers

**Affiliations:** 1Department of Respiratory Medicine, NUTRIM Institute of Nutrition and Translational Research in Metabolism, Maastricht University Medical Centre+, 6229 ER Maastricht, The Netherlands; 2Division of Clinical Physiology, St John’s Medical College & St John’s Research Institute, Bengaluru 560034, India; 3Department of Pulmonary Medicine, St John’s Medical College Hospital, Bengaluru 560034, India; 4Department of Health Promotion, NUTRIM Institute of Nutrition and Translational Research in Metabolism, 6229 ER Maastricht, The Netherlands; 5Department of Physiology, St John’s Medical College, Bengaluru 560034, India; 6Department of Health Promotion, Care and Public Health Research Institute (CAPHRI), Maastricht University, 6211 HX Maastricht, The Netherlands

**Keywords:** COPD, asthma, chronic respiratory disease, combined lifestyle intervention, management

## Abstract

(1) Background: A healthy lifestyle has a protective role against the onset and management of asthma and chronic obstructive pulmonary disease (COPD). Therefore, combined lifestyle interventions (CLIs) are a potentially valuable prevention approach. This review aims to provide an overview of existing CLIs for the prevention and management of asthma or COPD. (2) Methods: A systematic literature search was conducted using PubMed, EMBASE, and PsycInfo. Studies were included if CLIs targeted at least two lifestyle factors. (3) Results: Among the 56 included studies, 9 addressed asthma and 47 addressed COPD management, with no studies focusing on prevention. For both conditions, the most prevalent combination of lifestyle targets was diet and physical activity (PA), often combined with smoking cessation in COPD. The studied CLIs led to improvements in quality of life, respiratory symptoms, body mass index/weight, and exercise capacity. Behavioural changes were only measured in a limited number of studies and mainly showed improvements in dietary intake and PA level. (4) Conclusions: CLIs are effective within asthma and COPD management. Next to optimising the content and implementation of CLIs, these positive results warrant paying more attention to CLIs for persons with an increased risk profile for these chronic respiratory diseases.

## 1. Introduction

Chronic respiratory diseases (CRDs) are among the most common non-communicable diseases worldwide [1]. Two of the most prevalent CRDs are asthma and chronic obstructive pulmonary disease (COPD). The increasing prevalence of asthma and COPD is an alarming public health concern in low- and middle-income countries as well as in high-income countries [2,3]. In 2019, globally there were 262 million and 212 million cases of asthma and COPD, respectively, with COPD being the third leading cause of death [4]. While both asthma and COPD are treatable, they are not curable. Therefore, early diagnosis as well as prevention of onset and progression of disease are key.

Asthma and COPD are characterised by chronic airway inflammation leading to narrowing of the airways and respiratory symptoms including cough, wheezing, and dyspnea [5,6]. Respiratory impairment in COPD often co-exists with extra-pulmonary manifestations, adversely affecting disease progression and quality of life (QoL), including muscle wasting, bone loss, cardiovascular disease, anxiety, depression, and decreased cognitive function [7,8,9]. In asthma, common extra-pulmonary manifestations associated with a decreased QoL include overweight and psychopathologies [10]. Many of these manifestations are linked to lifestyle-related risk factors [11,12].

Smoking is a primary risk factor for COPD, but it is also an important risk factor for the development of asthma [13]. Additionally, a recent systematic review has emphasised the beneficial effects of physical activity (PA) on asthma control, lung function, and QoL [14]. Moreover, PA may reduce adult-onset asthma risk [15]. PA has also been shown to slow the progression of COPD and reduce COPD risk in active smokers [16]. Diet and nutrition have been shown to play a role in both the prevention as well as the progression of COPD [17]. An unhealthy Western eating pattern (high intakes of saturated fat, red and processed meat, and refined grains and low intakes of fruits, vegetables, whole grains, and fish) has been linked to COPD risk and a decline in lung function, while a diet rich in dietary fibre and certain nutrients, including vitamins C and E, polyphenols, and β-carotene, has been associated with a decreased risk of COPD [17]. A plant-based diet, rich in fruits and vegetables, may also provide protection against the development of asthma and a reduction in asthma symptoms [18]. Additionally, excessive alcohol use has been associated with pulmonary function decline and increased risk of asthma [17,19]. Furthermore, psychological stress and poor sleep quality have been associated with worse outcomes in asthma and COPD [20,21,22,23], making them important targets for intervention.

As a combination of various lifestyle factors may influence asthma and COPD onset and progression, a comprehensive approach (simultaneously) targeting behavioural changes in multiple lifestyle factors could be beneficial for high-risk groups and patients with asthma or COPD. This approach is also defined as a ‘combined lifestyle intervention’ (CLI). CLIs can be implemented by a combination of different healthcare professionals, including dieticians, physiotherapists, and psychologists, or by lifestyle coaches who are trained to coach individuals on changing various lifestyle factors. Advising, educating, training, and counselling are the methods often used in the implementation of CLIs, where counselling is the most effective method for increased and long-lasting behavioural change [24]. According to the official definition of pulmonary rehabilitation (PR), a PR programme can be considered a CLI [25]. However, in many PR programmes, exercise training is still the cornerstone and less attention is given to other lifestyle factors [26].

There is growing interest in the effectiveness of CLIs worldwide, particularly in the management of overweight and obesity for the prevention of cardiometabolic diseases [27,28]. However, the extent of existing CLIs for the prevention and management of CRDs, as well as evidence of their effects, have not been systematically investigated and documented. The aim of this systematic review is, therefore, to give an overview of existing CLIs that target a behavioural change in at least two lifestyle factors, including PA level, dietary intake, smoking behaviour, alcohol consumption, stress level, or sleeping behaviour, for the prevention and management of asthma or COPD in adults. Additionally, this review aims to assess the effects of these CLIs on behavioural and physiological outcomes.

## 2. Materials and Methods

This systematic review is reported in accordance with PRISMA guidelines [29] and pre-registered in the PROSPERO database (CRD42022382321) [30]. Detailed methods are presented in the Appendix A.

### 2.1. Data Source and Search Strategy

A systematic literature search was conducted in MEDLINE via PubMed, EMBASE via OVID, and PsycInfo. The initial search was performed on 22 June 2022 and repeated on 24 August 2023 to check whether new articles had been published in the past year. In order to obtain a complete overview of the available literature, no publication date limit was implemented. The search strategy included relevant terms for the population, intervention, and study design (PICOS) (Appendix A). Details on the search strategy and the search strategy for each database are presented in the Appendix A.

### 2.2. Study Selection and Eligibility Criteria

Articles obtained with the search were first screened based on the title and abstract for relevance, after which relevant full-text articles were assessed for eligibility. Two researchers (C.D.C. Born and R. Bhadra) independently performed the screening of the obtained articles in Rayyan [31]. Studies that targeted the prevention and management of asthma or COPD, and as such included patients diagnosed with asthma or COPD or individuals at high risk of developing asthma or COPD (> 18 years old), were eligible for inclusion (Appendix A). Additionally, articles were considered for inclusion when they reported on CLIs that targeted a behavioural change in at least two of the following lifestyle factors: PA level, dietary intake, smoking behaviour, alcohol consumption, stress level, or sleeping behaviour. Moreover, in the cases of RCTs and non-randomised controlled trial studies, the intervention group had to differ from the control group in at least two lifestyle factors being targeted. Studies were excluded when they focused on environmental changes to modify behaviour (e.g., constructing more bicycle lanes to stimulate PA), included CLIs in combination with pharmacological treatment, primarily focused on the effects of nutritional supplementation or physical exercise training, and when they were conducted in an inpatient setting. The CLIs needed to be delivered human-to-human, via physical or digital contact, and could be implemented either as individual or group interventions. Only publications in the English language and research articles for which results were available were included. Single group pre–post studies, RCTs, and non-randomised controlled trial studies were eligible for inclusion. In the cases of RCTs and non-randomised controlled trial studies, between group differences needed to be reported.

### 2.3. Data Extraction and Synthesis

Data extraction was performed by two reviewers (C.D.C. Born and R. Bhadra) who crosschecked each other’s extraction (data extraction form available on https://osf.io/vw9ej). See the Appendix A for a detailed description of the extracted data. The data were analysed separately based on the reported CRD. Effects of the CLIs on outcomes were presented in a table using colour coding. The colour green was used to signify a significant desired effect on the outcome, grey was used to signify no change in outcome (non-significant effects), and red was used to signify a significant undesired effect on the outcome. Desired effects were interpreted as improvements in the reported outcomes from a health perspective, which was sometimes dependent on the study population and aim of the specific study (e.g., weight gain could be desired in patients who were underweight while weight loss could be desired in patients who were overweight). Regarding the time points for which the outcomes were reported, the first measurement after the end of the intervention was reported, as well as all additional follow-up measurements. For RCTs and non-randomised controlled clinical trials, between group effects were reported, and for single group pre–post studies, within group differences were reported.

### 2.4. Risk of Bias (Quality) Assessment

The quality of the included studies was assessed using the Quality Assessment Tool for Quantitative Studies, developed by the Effective Public Health Practice Project at McMaster University [32]. This tool was used to evaluate the quality (indicated as “strong”, “moderate”, or “weak”) of selected studies in eight different domains. See the Appendix A for a detailed description of the quality assessment.

## 3. Results

### 3.1. Study Selection

A total of 4339 research articles were identified from the various databases and three from citation searching; subsequently, 3318 articles were screened based on the title and abstract after removing duplicate records (Figure 1). Two hundred forty-seven potentially eligible articles were considered for full-text screening, of which full texts of three articles were not retrieved after approaching the authors. The full texts of 244 articles were reviewed, and 71 articles describing 56 intervention studies were included based on the predefined inclusion and exclusion criteria. All three articles identified from citation searching [33,34,35] were identified from protocols found with the search [36,37,38].

The characteristics of the 56 quantitative studies are summarised in Table 1. In addition, four qualitative studies were identified, which were embedded into large RCTs conducted among patients with COPD, are included in Table 1 [39,40,41,42]. These qualitative studies described patients’ experiences of participating in the intervention trials and the impact of the interventions on behavioural outcomes. The most frequently reported changes in health behaviour were increased PA, reported by all four studies, followed by improved dietary intake reported in two studies, and one study reported perceived improvement in smoking behaviour. Twelve protocols fit the inclusion criteria, nine of which the actual studies were included in the review [36,37,38,43,44,45,46,47,48] and three of which the results were not available yet and were therefore excluded [49,50,51].

The systematic review of the literature only provided articles describing CLIs for the management of asthma and COPD. No CLIs in high-risk groups to prevent the onset of asthma or COPD were obtained.

### 3.2. COPD

#### 3.2.1. Study Characteristics

A total of 47 studies were included, of which 26 were RCTs [33,35,52,53,54,55,56,57,58,59,60,61,62,63,64,65,66,67,68,69,70,71,72,73,74,75], 3 were non-randomised controlled trials [76,77,78], and 18 were single group pre–post studies [34,79,80,81,82,83,84,85,86,87,88,89,90,91,92,93,94,95]. The methodological quality of the studies ranged from strong to weak, with the majority being moderate and strong (Appendix A, quality assessment table available on https://osf.io/2jcm7). Quality seemed to be independent of study design; RCTs of both high and low quality were included. There was heterogeneity in the number of recruited participants across the studies, which ranged from 11 to 8217 participants (Table 1). The mean age of the participants ranged between 56.2 years to 75.0 years and the mean body mass index (BMI) ranged from 17.8 kg/m^2^ to 36.1 kg/m^2^, but there was only one study including participants with a mean BMI > 30 kg/m^2^ [86]. A higher male to female ratio was reported in the majority of the studies. The percentage of current smokers varied widely across the studies, ranging from 5.5% to 69.2%. Based on the mean levels of forced expiratory volume in 1 s (FEV1) reported across the studies, the majority of the patients had moderate COPD according to the Global Initiative for Chronic Obstructive Lung Disease (GOLD) [6]. The majority of the studies reported on interventions with a duration of 8–12 weeks [52,53,56,57,60,69,72,73,77,78,79,80,81,82,83,84,86,87,88,90,91,93,94,95], but also very short-term (3–6 weeks [54,62]) and longer term (i.e., 4 months to 1 year [34,35,55,59,61,64,65,67,70,71,74,75,85,89] and extending beyond 1 year [58,63,66]) interventions were included. Sixteen studies (34%) investigated a pulmonary rehabilitation programme that targeted more than one lifestyle factor [52,54,58,60,65,74,77,78,79,80,81,82,83,88,93,94].

As shown in Table 1 and Appendix A, the majority of studies were implemented in Europe (*n* = 19, 40%), followed by Asian countries (*n* = 9, 19%), the USA (*n* = 8, 17%), and Australia (*n* = 6, 13%). Other studies were conducted in Canada (*n* = 1, 2%), Brazil (*n* = 2, 4%), and South Africa (*n* = 1, 2%). For one study, the location of implementation was not specified [94].

#### 3.2.2. Investigated Lifestyle Factors and Reported Outcomes

##### Lifestyle Factor Targets in CLIs

Forty-five CLIs were described across the 47 included studies. Bourne et al., 2022 and Mitchell et al., 2014 studied the same CLI, but with a difference in the mode of delivery (group-based approach versus individual level, respectively). Furthermore, Benzo et al., 2013 and Benzo et al., 2016 studied the same CLI, the former in a pilot study with a smaller study population. The majority of CLIs targeted a combination of three lifestyle factors (*n* = 23, 49%; Table 2) [35,52,55,57,58,59,62,63,69,73,74,75,76,77,78,79,83,84,85,87,88,90,91,92]. Furthermore, there were 15 (32%) CLIs focusing on a combination of two lifestyle targets [33,34,53,60,64,66,68,70,71,72,80,81,86,93,95], five (11%) focusing on a combination of four lifestyle targets [54,65,67,82,94], and three (6%) focusing on a combination of five lifestyle targets [56,61,89]. The highest number of CLIs targeted the combination of diet + PA + smoking (*n* = 14, 30%), followed by the combination of PA + smoking (*n* = 8, 17%; Figure 2 & Table 2). Diet and PA were, however, the most frequently occurring lifestyle targets across all CLIs (*n* = 32, 68%). From these studies, five (16%) CLIs targeted the combination of just diet and PA [60,80,81,86,93], while in 27 (84%) studies, they were targeted in combination with additional lifestyle factors [35,52,54,55,56,57,58,59,61,62,63,65,67,73,75,76,77,78,79,82,83,85,87,88,89,90,94]. In 21 (78%) of these 27 studies, smoking cessation was included [35,52,55,56,57,58,59,61,63,65,67,73,76,78,82,83,85,87,89,90,94], and in 12 (44%) studies, stress management was targeted [54,56,61,62,65,75,77,79,82,88,89,94]. Except for one study [92], stress management always occurred in combination with targeting PA (*n* = 16, 34%). Six combinations of lifestyle factor targets, including (1) PA + smoking + alcohol, (2) diet + smoking + sleep, (3) diet + smoking + stress management, (4) diet + PA + stress management + sleep, (5) diet + PA + smoking + sleep, and (6) diet + PA + smoking + alcohol + stress management, only occurred once [54,61,67,74,91,92]. Sleeping behaviour and alcohol consumption were targeted least frequently in combination with other lifestyle factors, in five (11%) [54,56,67,89,91] and two (4%) CLIs [61,74], respectively.

#### 3.2.3. Reported Outcomes and Possible Effects of CLIs

##### Behavioural Outcomes

Only two (4%) studies reported on eating behaviour and both found improvements, described as a decrease in total energy intake and increase in protein intake [86] and an increase in eating a balanced diet [76] (Figure 3A). Nine (19%) studies reported on PA levels, of which five found an increase in PA level [64,67,70,76,96] and four found no effect [63,65,69,95]. Out of 10 (21%) studies reporting an outcome on smoking cessation, only one found a decrease in the number of current smokers [66], while the other nine studies did not find any effect [35,59,63,70,71,76,89,90,95]. No studies measured the effects on alcohol consumption or sleeping behaviour and only one study reported a decrease in stress level [94].

##### Physiological Outcomes

The number of studies reporting physiological outcomes (*n* = 35, 74%) was higher than that reporting behavioural outcomes (*n* = 16, 34%; Table 2). Fifteen (32%) studies reported on lung function (FEV1), of which nine found no effect [35,53,55,59,64,71,82,83,86] and six reported a desired effect (Figure 3A). In five cases, this was an increase in FEV1 level [73,81,87,88,93], and in one case, this was a less severe decrease in FEV1 level compared to the control group [66]. The majority of studies reporting the effects on dyspnoea reported an improvement (*n* = 9) [60,66,73,78,80,83,91,93,94], while six studies found no change in dyspnoea [63,70,76,85,89,95]. One of these studies reported the effects separately for ex-smokers and current smokers, and it only found an improvement in dyspnoea in current-smokers [83]. Of the seven (15%) studies reporting on BMI/weight [58,60,65,66,83,86,93], four found a desired effect. In three studies, this was an increase in weight [58,60,66], and in one study, including patients with obesity and COPD, this was a decrease in weight [86]. One of the studies that did not find an effect on BMI did report improvements in body composition (improved fat free mass and reduced fat mass) [93]. Twenty-three (49%) studies reported the effect on exercise capacity, with 16 showing an improvement [52,54,60,62,64,66,73,78,79,81,82,83,86,89,93,94] and seven showing no effects [57,58,65,71,75,77,87]. Three studies (6%) measured muscle strength [57,58,93], of which two showed an improvement [58,93].

##### Quality of Life

QoL was reported in 89% (*n* = 42) of the studies (Table 2). Out of these 42 studies, 23 found a desired effect [34,53,54,55,62,68,69,74,75,78,79,80,82,83,84,86,87,88,89,91,92,93,94] and 19 found no effect [33,35,52,56,57,58,59,61,63,64,65,67,70,71,72,76,77,85,95] (Figure 3A).

##### Follow-Up Measurements

Thirteen studies also conducted follow-up measurements after completion of the intervention. In three studies, the CLI included a maintenance phase in which participants still received intervention components, but on a less frequent basis, to encourage adherence to the targeted lifestyle behaviours [58,65,73]. Follow-up measurements after the maintenance phase remained the same compared to the effects measured at the end of the intervention, only the beneficial effects on weight were not maintained after 24 months in the study by van Wetering et al., 2010 (Table 2). The other studies did not include a maintenance phase but did report follow-up outcome measurements [52,54,56,62,68,69,70,75,91,94] (See Table 2 for follow-up durations). Most effects on the reported outcomes remained the same after follow-up. However, in some studies, the effects after follow-up were no longer detectable on exercise capacity [54], PA level [70], stress level, and symptoms [94]. One study found no effect on QoL after the end of the intervention (12 weeks) but did find a desired effect after follow-up (24 weeks) [52].

### 3.3. Asthma

#### 3.3.1. Study Characteristics

A total of nine studies were identified, including six RCTs [97,98,99,100,101,102] and three single group pre–post studies [103,104,105] (Table 1). The number of participants across the studies ranged from 27 to 330 participants. The methodological quality varied between the studies (strong *n* = 2 [103,104], moderate *n* = 3 [100,101,102], and weak *n* = 4 [97,98,99,105]) (Appendix A). The mean age of the participants across the studies ranged from 33.4 years to 55.0 years. The mean BMI across the studies ranged from normal weight to obese (from 22.6 to 37.6 kg/m^2^). The majority of the studies reported a higher female to male ratio and ≤7% of current smokers, except one study that had 36.7% current smokers [104]. The FEV1 levels ranged from 62.5% to 82.6% percentage predicted volume. The intervention duration in five of the studies was 8–12 weeks [97,99,101,102,103]. There was one study with a 7-week duration [98], two studies had a duration of six months [104,105], and one study had a duration of 12 months [100].

Five (56%) studies were conducted in the USA, two were conducted in Europe, one was conducted in Saudi Arabia, and one was conducted in India (Table 1 and Appendix A).

#### 3.3.2. Investigated Lifestyle Factors and Reported Outcomes

##### Lifestyle Factor Targets in CLIs

The studies that described CLIs for the management of asthma targeted a combination of two (*n* = 6, 67%) [98,99,100,101,102,104] and three (*n* = 3, 33%) lifestyle factors [97,103,105] (Table 2). Again, the combination of diet and PA as lifestyle targets occurred most frequently across all studies. The combination diet + PA was targeted in the highest number of studies (*n* = 3, 33%) [100,101,102] (Figure 2). Moreover, three (33%) studies targeted diet and PA in combination with another lifestyle factor, namely smoking (*n* = 1) [103], alcohol consumption (*n* = 1) [105], and stress management (*n* = 1) [97]. The lifestyle target combinations of (1) PA + smoking, (2) PA + SM, and (3) diet + stress management also only occurred once [98,99,104].

#### 3.3.3. Reported Outcomes and Possible Effects of CLIs

##### Behavioural Outcomes

Two (22%) studies reported desired effects on dietary intake, described as a reduction in total caloric, fat, and carbohydrate intake and an increase in the percentage of protein in the participants’ diets [105] and an increase in protein intake and reduction in the dietary glycaemic index [101] (Figure 3B). Two (22%) studies reported on PA levels [98,100], of which one showed an increase in PA level [98]. One study reported on stress levels but found no effect [98]. No studies reported any outcomes on smoking behaviour, alcohol consumption, or sleeping behaviour.

##### Physiological Outcomes

There was a higher number of studies reporting physiological outcomes (*n* = 8, 89%) [97,99,100,101,102,103,104,105] than behavioural outcomes (*n* = 4, 44%) [98,100,101,105] (Table 2). Of seven (78%) studies reporting the effect on lung function (FEV1) [97,99,100,101,103,104,105], three found an improvement [97,99,104] (Figure 3B). Four [99,101,103,104] out of six (67%) [99,100,101,103,104,105] studies reporting an effect on respiratory symptoms found improvements. Five studies (56%) reported on BMI/weight, and only one study found no effect [103], while the other four studies found a decrease in weight [100,101,102,105]. None of the studies reported outcomes related to exercise capacity or muscle strength.

##### Quality of Life

Besides lung function, QoL was the most frequently reported outcome, reported in seven (78%) studies. The majority of studies evaluating the effect on QoL found an improvement (*n* = 6) [97,98,99,101,103,105], and only one study found no effect [100] (Figure 3B).

##### Follow-Up Measurements

Two studies conducted follow-up measurements after completion of the intervention (Table 2). In the study by Pokladnikova et al., 2013 [99], the effects after six months stayed the same as those measured after 2 months. Bentzon et al., 2019 [106] could no longer find an improvement in respiratory symptoms at the one-year follow-up, which was detected in the EFFORT trial by Toennesen et al., 2018 [101] after an 8-week intervention.

**Table 1 nutrients-16-01515-t001:** Descriptive table including details of the included studies and the described combined lifestyle interventions. Studies are reported based on the number of lifestyle factors targeted (high to low) and ordered according to study design. A more detailed version of the descriptive table, including patient characteristics, is provided in the Appendix A.

Authors, Year, Reference	Country of Implementation CLI	Study Design	Participants, N	Intervention Details	Control Group
**COPD**					
**Kheirabadi 2008 [56]**	Iran, Isfahan	Randomised controlled trial	Active: 21 Control: 21	**Duration**: **8 weeks** Eight 60–90 min educational sessions with a 1-week interval in 3–4 member groups: - Educational sessions on the disease, drug use, respiratory techniques and use of a self-management programme. - Behavioural modification focused on a healthy lifestyle, smoking cessation, healthy sleep, nutritional habits, stress management, free time activities, simple regular exercise programme at home, and behavioural interventions focusing on common issues like independence and self-esteem.	Usual care—not further specified
**Walters 2013 [61]**	Australia, Tasmania	Randomised controlled trial	Active: 90 Control: 92	**Duration: 12 months** 16 telephone sessions of 30 min with increased time between calls. The intervention consisted of cognitive behavioural health mentoring. Participants set medium-term to long-term goals using a specified framework of health behaviour targets, namely smoking, nutrition, alcohol, physical activity, psychosocial wellbeing, and symptom management. Individualised ‘action’ plans were set up to reach their goals and there was constant review and revision of the action plan.	Received their usual care as provided by a GP plus regular monthly phone calls from a research nurse. The telephone calls did not provide specific psychological advice or skills training.
**Wilson 2015 [65]**	UK, Norfolk	Randomised controlled trial	Active: 73 Control: 75	**Duration: 1 year** **Maintenance PR programme** (after 8 weeks of outpatient PR): 2 h session every 3 months, including 1 h of education and 1 h of exercise training. This was supervised tailored exercise training followed by a home-based exercise training prescription. Education sessions covered smoking cessation, healthy eating, exercise importance, and coping and dealing with psychological issues. Participants received an invitation to attend Norwich Breathe Easy Group (provides support and advice for those living with lung disease).	Standard care, advice to exercise at home and an invitation to attend Norwich Breathe Easy Group (provides support and advice for those living with lung disease.) Patients underwent an outpatient PR programme for 8 weeks before randomisation.
**Jonsdottir 2015 [67]**	Iceland, Reykjavik	Randomised controlled trial	Active: 48 Control: 52	**Duration: 6 months** Partnership-based **self-management programme**: - 3–4 patient/family conversations (30–45 min) with a clinical nurse specialist covering main health concerns, symptoms, daily life impact, disease nature and management, smoking cessation, emotional challenges associated with the disease, and possibilities for patient/family to prevent further decline of disease. - Smoking cessation treatment: ≥1 face-to-face sessions with a clinical nurse specialist followed by ≥3 telephone or face-to face conversations combining psychosocial support with pharmacological treatment. - Group meetings lasting 2 h featuring research team presentations, provision of written educational material, COPD volunteer talks, and group discussions (5–15 patient/family members). Focus on quitting smoking assistance, wellbeing activities (exercise, sleep-rest-activity-relaxation, nutrition, anxiety, and depression).	Traditional health care: services provided by general practitioners at primary healthcare centres and visits to lung physicians based on referral from general practitioners or self-initiated appointments.
**Finnerty 2001 [54]**	UK, not further specified	Randomised controlled trial	Active: 36 Control: 29	**Duration**: **6 weeks** **Outpatient PR** included 2 weekly visits: a 2 h education visit and a 1 h exercise visit. The intervention was conducted by: - Physiotherapist: assisted aerobic activity exercises with advice for daily home exercises (1–2 times, ≥5 times/week). - Dietician: dietary assessment and advice, weekly monitoring progress in recommended eating habits. - Occupational therapist: consultation once weekly for the first 4 weeks. - Respiratory specialist nurse: counselling on anxiety management and relaxation techniques in week 2, sleep problems in week 5, and discussion on relaxation techniques in week 6. Participants were also invited to attend the monthly local “Breathe Easy” club and subsequent “drop-in” exercise sessions at weeks 8, 9, and 10.	Patients reviewed routinely as medical outpatients
**Bendstrup 1997 [52]**	Denmark, Esbjerg	Randomised controlled trial	Active: 16 Control: 16	**Duration: 12 weeks** **Outpatient PR:** - Physiotherapist guided 1 h of exercise training thrice weekly for 12 weeks, with emphasis on home training continuation. - 12 educational sessions optimising disease management, also featuring a psychologist, social worker, and dietitian (one session each). - Smoking cessation intervention by a physician and occupational therapist, offering free nicotine patches. Initial programme introduction covered benefits and challenges followed by two reinforcement sessions. A folder with advice was also available. Practical alternatives to smoking were provided.	Details not specified
**Dheda 2004 [55]**	UK, London	Randomised controlled trial	Active: 10 Control: 15	**Duration: 6 months** Outpatient follow-up with a respiratory nurse and/or chest physician ≥ 4 times within 6 months (at 3, 6, 8, 12, and 16 weeks). Interventions at these visits included: smoking cessation advice, nutrition and exercise advice, and introduction to a patient support group.	Visited primary care team on “need to” basis
**Theander 2009 [57]**	Sweden, not further specified	Randomised controlled trial	Active: 12 Control: 14	**Duration: 12 weeks** Multidisciplinary team intervention, comprising a physiotherapist, dietician, occupational therapist, and a nurse, involving: - Two weekly 1 h physiotherapy sessions (aerobic and strength training). After 1 month, individualised home training was added. - Individual dietary advice addressing energy and nutrient intake. Three 1 h sessions occurred at weeks 2, 4, and 8, with additional nutritional supplementation for patients with a BMI < 20 and special advice for patients with a BMI > 30. - Smoking cessation advice sessions at weeks 2 and 4.	The control group did not receive any of the multidisciplinary rehabilitation programme or care from the multidisciplinary professionals who performed the rehabilitation programme.
**van Wetering 2010 [58]**	The Netherlands, not further specified	Randomised controlled trial	Total (patients with muscle wasting) Active: 16 Control: 23	**Duration: 4-month rehabilitation** phase and 20-month maintenance phase The **INTERCOM trial** (4-month PR phase) offered by local dieticians, physiotherapists, and respiratory nurses included: - Two weekly 30 min physiotherapist visits for supervised exercise training. - Counselling session (at intervention start and after 1, 2, and 4 months) by a dietician and a standardised nutritional therapy: 3 oral liquid supplements per 24 h (564 kcal). - Individualised education and standardised smoking cessation counselling by a respiratory nurse. The 20-month maintenance phase included: - Monthly physiotherapist visits - 4 dietician consultations (after 6, 9, 12, and 24 months), with continued nutritional supplements as needed.	The usual care group received pharmacotherapy according to accepted guidelines, a short smoking cessation advice by their chest physician, and if they met the criteria for nutritional support, a verbal recommendation to improve dietary intake.
**Zwar 2012 [59]**	Australia, Sydney	Randomised controlled trial	Active: 234 Control: 217	**Duration: 6 months** Patients had 2 home visits and 5 telephone contacts with a nurse and a minimum of two GP consultations. The nurse and GP met face-to-face on two occasions followed by monthly or more frequent telephone consultations as needed to discuss progress and issues. Using nurse assessments and patient discussions (goal setting and action planning), individualised care plans were created covering smoking cessation, PR, nutrition, psychosocial matters, patient education, COPD comorbidities, and complications.	Patients received usual care, which was defined as processes normally followed by the GP and the patient regarding review, pharmacological therapy, and management of COPD.
**Kruis 2014 [63]**	The Netherlands (western part)	(Cluster) Randomised controlled trial	Active: 554 Control: 532	**Duration: 24 months** **The RECODE trial** implemented the intervention at the cluster level. GPs, PNs, and physiotherapists underwent 2-day training on integrated disease management. Patients in the intervention group received integrated disease management implemented by the multidisciplinary team consisting of at least three members: the GP, the practice nurse, and physiotherapist. Depending on the team needs, a pulmonary physician and dietician were added to the team. Essential components of COPD-disease management included: motivational interviewing, smoking cessation counselling, physical (re)activation, and nutritional support. The intensity of the integrated disease management programme for individual patients depended on their health status, personal needs, and preferences, as well as on the capacity of the general practice team.	The control group continued usual care (based on the 2007 national primary care COPD guidelines).
**Zwar 2016 [35]**	Australia, Sydney	(Cluster) Randomised controlled trial	Active: 144 Control: 110	**Duration: 12 months** Nurses and GPs attended workshops for team-based COPD management education. PNs collaborated with GPs and patients to create care plans for patients newly diagnosed with COPD. Workbooks from the training guided them to include the following components: - Smoking cessation. - Exercise: recommended for all, regardless of COPD severity. - Nutrition: advice about diet and exercise and dietician referral for appropriate cases. - Psychosocial issues: identification and anxiety/depression management. - Assessment and management of co-morbidities and complications of COPD. - Patient education: written information about COPD (management) and local support groups for emotional support and self-management.	After case-finding training for nurses, staff in control practices received no further intervention other than GPs in these practices being mailed a copy of the COPD-X guidelines (national guidelines).
**Zhang 2020 [73]**	China, Zunyi	Randomised controlled trial	Active: 85 Control: 89	**Duration: 3 months** - Physical exercise (40–55 min, twice a week) led by a physiotherapist, Tai chi mentor, and respiratory nurse. - Smoking cessation: 2 sessions (group and individual) with a psychologist and respiratory nurse. - Self-management education every 2 weeks by a physician, respiratory nurse, and nutritionist on: COPD knowledge, home exercise, symptom management, medication instruction, lifestyle change, and nutrition support. - Psychosocial support: 2 sessions with a psychologist and respiratory nurse, including a group activity for communication among patients, and education on emotional coping. Post intervention, a 24-month follow-up by a respiratory nurse ensured continuous care with individualised home exercise prescribed. Periodical telephone follow-up (once every 1–2 weeks) and home visits (once every 1–3 months) to record and ensure home exercise and smoking adherence.	Usual care: discharge education about self-management, exercise training, medication, and seeking health care when necessary. Each patient in this group got a pamphlet addressing self-management of COPD, including symptom recognition, smoking cessation, physical exercise, medication use, oxygen therapy, and nutrition. Contact information was printed in the pamphlet for health counselling service.
**Zhu 2021 [74]**	China, Jiangsu	Randomised controlled trial	Active: 40 Control: 41	**Duration**: **6 months** Community-based **rehabilitation** with outpatient rehabilitative treatment tailored to age, physical conditions, and COPD severity. - Rehabilitation education: lectures promoting smoking and alcohol cessation and fostering healthy habits. - Personalised exercise plans including regular walking, cycling, and treadmill use. - Psychological counselling: provided by psychologists for patients with negative emotions like anxiety, depression, and boredom.	Regular outpatient rehabilitative treatment—not further specified
**Mitchell 2014 [62]**	UK, Leicester	Randomised controlled trial	Active: 89 Control: 95	**Duration**: **6 weeks** **Self-management program by a** physiotherapist: - SPACE for COPD manual for home use with educational material (“How to get/stay fit”, “Managing your stress”, “Healthy eating”). - Home exercise programme: Introduction in a 30–45 min consultation advising on manual use and explaining the exercise regime. Participants receive telephone calls at 2 and 4 weeks to reinforce skills and encourage progress using goal-setting and motivational interviewing techniques.	Usual care—not further specified
**Benzo 2016 [69]**	USA, Minnesota	Randomised controlled trial	Active: 108 Control: 107	**Duration: 8 weeks** Health coaching intervention (same as Benzo et al. (2013) [84]) including 8 weekly counselling sessions emphasising self-management via motivational interviewing principles: 1 face-to-face session (±2 h), followed by 7 scheduled telephone sessions. Key action plan domains: smoking cessation, coping, stress management, and increasing physical activity. Patients received the book “Living a Healthy Life with Chronic Conditions”. Patients were provided with a Stamina In Motion Elliptical Trainer for daily seated use (aiming for 20 min/day) and instructed on three simple upper extremity exercises (five repetitions) from the book.	Usual care and referral for PR
**Gurgun 2013 [60]**	Turkey, Izmir	Randomised controlled trial	Active: 15 Control: 16	**Duration**: **8 weeks** **PR + Nutritional intervention**: - PR: education (not specified) and exercise training (60–80 min/day twice weekly). - Nutritional intervention: patients received three 250 mL nutritional drinks daily (83.3% carbohydrates, 30% fat, and 16.7% proteins) and were encouraged to continue regular meal portions. Additionally, dietary advice was provided.	Usual care—not further specified
**Lou 2015 [66]**	China, rural areas of Xuzhou City	Randomised controlled trial	Active: 4197 Control: 4020	**Duration**: **4 years** - GPs received 2 days of health management training for the intervention: Designing a health management plan - 40–60 min lecture every 2 weeks, focusing on smoking cessation counselling, encouragement for regular exercise, rehabilitation, and psychological counselling. Subjects received educational materials covering 48 lectures. - Every 2 weeks, at least one face-to-face follow-up visit to assess compliance by the GP. Monthly reports on subjects were submitted to a team of professionals (pulmonologists, psychiatrists, rehabilitation specialists, nutritionists, and respiratory nurses) for assessment. The team selected a focus topic for each subject, and assessments were returned to the GPs for continued improvement of the follow-up procedure.	Received usual care from healthcare providers and GPs. Follow up telephone/face-to-face visits every 2 months. Medical management and referral to respiratory specialist when indicated. Content and frequency not standardised.
**Suhaj 2016 [68]**	India, Manipal	Randomised controlled trial	Active: 130 Control: 130	**Duration: not specified** Education by trained pharmacists (one-on-one, 15–20 min) and distribution of patient information leaflets (PILs) reinforcing counselling content, emphasising smoking cessation, and simple exercise. Two years of patient follow-up included monthly telephone calls to ensure medication adherence and timely follow-ups.	Standard hospital care
**Markun 2018 [33]**	Switzerland, Zurich	Randomised controlled trial	Active: 101 Control: 115	**Duration: Not specified** The intervention involved a half-day workshop for GPs including a knowledge refreshment on Swiss COPD guidelines, distribution of pocket guides, and a discussion with GPs and practice assistants on tailoring individual COPD care pathways. GPs were asked to use the COPD care bundle as a checklist for key elements, boosting internal motivation for behavioural change. After 6 months, a 3 h refresher workshop was conducted for the practice teams. Key elements of COPD care included: smoking cessation advice and intervention, assessment and advice on physical activity, patient education class referral, integration of other healthcare providers, and referral to pulmonary rehabilitation.	No intervention delivered to the “usual care” control group
**Jolly 2018 [70]**	England, 71 general practices located at Birmingham and West Midlands South, Greater Manchester, West Midlands North, and Oxfordshire or Gloucestershire	Randomised controlled trial	Active: 289 Control: 288	**Duration: 24 weeks** **Telephone health coaching** by nurses with supporting documents, a pedometer, and a self-monitoring diary. The intervention included education, monitoring, and assessment in order to increase self-efficacy and followed Social Cognitive Theory. Content: smoking cessation, physical activity increases, correct inhaler use technique, and medication adherence. The first coaching session was within 1 week of randomisation, lasting 35–60 min. Follow up telephonic sessions were at weeks 3, 7, and 11 (15–20 min), with standard prompts or information at weeks 16 and 24.	Usual care + 13 page standard information leaflet about self-management of COPD.
**Thom 2018 [71]**	USA, San Francisco	Randomised controlled trial	Active: 100 Control: 92	**Duration: 9 months** **Health coaching** by trained coaches for patient self-management. Coaches aimed for an initial visit within 2–3 weeks of enrolment and ≥3 in-person meetings during the study. Phone check-ins occurred at least every 3 weeks, including within 2 weeks after each medical visit (≥13 telephone check-ins over 9 months). Additional contacts were guided by patient needs and preferences. Content: enhancing disease understanding and symptom awareness, improving use of inhalers, making personalised plans to increase physical activity, smoking cessation, or otherwise improve disease management, and facilitating care coordination. Coaches, unlicensed health workers, underwent ±100 h of training over 3 months.	Usual care, included regular physician consultation and any other resources offered by their provider or clinic, i.e., access to COPD educators, respiratory therapists, COPD education classes, pulmonary rehabilitation, smoking cessation classes, and pulmonary specialist referrals by the primary care clinician.
**Aboumatar 2019 [72]**	USA, Baltimore, Maryland	Randomised controlled trial	Active: 120 Control: 120	**Duration: 3 months** Delivered by COPD nurses who met with the patient (and caregiver whenever possible) during hospital stay and post-hospital stay (up to 3 months), providing self-management support and addressed barriers to care: - Transition support for discharge preparation and understanding post-discharge care. - Individualised COPD self-management: medication guidance, recognising exacerbations signs, following an action plan, practicing breathing exercises, energy conservation, maintain an active lifestyle, seeking help, and smoking cessation. - Facilitated access to community programmes and treatment services.	Usual transitional care provided at the study site. This included assigning a general transition coach to follow up the patient for 30 days after discharge, focusing on adherence to the discharge plan, and connecting to outpatient care.
**Emery 1998 [53]**	USA, not further specified	Randomised controlled trial	Active: 29 Control: 25	**Duration: 10 weeks** **Exercise, education, and stress management (EXESM)** programme comprised 37 exercise sessions, 16 educational sessions, and 10 stress management classes. Participants met daily for ±4 h over 5 weeks, with daily 45 min exercise training, weekly 4 h COPD educational sessions, and 1 h of stress management and psychological support by a clinical psychologist driven by cognitive behavioural therapy. After the initial intensive 5 weeks, participants participated in a less intense 5-week regimen: exercise sessions thrice weekly for 60–90 min and 1 h weekly stress management classes.	Waiting list: were advised not to significantly alter activities
**Blumenthal 2014 [64]**	USA, North Carolina and Ohio	Randomised controlled trial	Active: 162 Control: 164	**Duration: 16 weeks** **Coping Skills Training (CST):** Patients and partners received telephone counselling on cognitive behavioural coping by a clinical psychologists weekly for 12 weeks and biweekly for 1 month (total 14 sessions of 30 min). The components of CST included stress education, coping skills training, individualised exercise prescription, and maintenance/generalisation.	COPD education via 12 weekly and 2 biweekly calls from a health educator covered topics like pulmonary physiology, medication usage, nutrition, and symptom management. Coping strategies were not addressed.
**Bourne 2022 [75]**	UK, Leicester	Randomised controlled trial	Active: 97 Control: 96	**Duration: 5 months** Participants received a SPACE for COPD manual (same as Mitchell et al., 2014 [52]) and attended the SPACE for COPD group-based (up to 10 participants) self-management programme facilitated by two trained healthcare professionals (HCPs) (e.g., physiotherapist, respiratory specialist nurse, occupational therapist, health psychologist). The programme, delivered in six 2 h sessions, over 5 months, included various topics such as medication, breathing control, exercise, and nutrition. Earlier sessions were delivered closer together in time. 12 HCPs attended a 1-day training and received an HCP delivery manual. Session content (all included goal setting): - Introduction to SPACE for COPD: Educational topics on medication, breathing control, exercise, and nutritional advice - Introducing exercise and managing shortness of breath–including introduction to the walking programme. - Continuing exercise and saving energy, including strength training. - Managing stress and emotions and the COPD action plan. - Keeping going from here, including maintaining exercise. Participants completed the exercise component of at home.	Participants in the control group continued with any usual check-ups/reviews—no additional care was provided or removed from their current access. If participants were referred to PR in the duration of their time in the study, they were not denied access to the programme. No additional advice, information or recommendations were provided to participants in this group.
**Monteagudo 2013 [76]**	Spain, Barcelona	Non-randomised controlled trial	Active: 400 Control: 401	**Duration: not specified** A programme consisting of an education and motivation workshop for 64 healthcare professionals (32 clinicians and 32 nurses). The 20 h workshop covered COPD guidelines, motivational interviewing, smoking cessation, inhaler use, diet counselling, exercise, physiotherapy, and audit/patient feedback.	Healthcare professionals in the control group did not participate in the workshop and followed standard clinical care.
**Da Silva 2018 [78]**	Brazil, Fortaleza	Non-randomised controlled clinical trial	Active: 38 Control: 36	**Duration: 12 weeks** Outpatient **PR** (3 sessions/week, 60 min each) by a multidisciplinary team (physiotherapist, chest physician, dietician, occupational therapist, psychologist, and social worker). Physical training included upper/lower limb stretches and strength and endurance training. Psychosocial team provided nutritional support, psychological counselling, education on COPD, smoking cessation, exacerbations, respiratory medication, and physical activity importance.	Patients in a waiting list awaiting admission to PR. Received medical management and were informed about the importance of physical activity; not followed by multidisciplinary team.
**Zakrisson 2011 [77]**	Sweden, multi-centre study, multiple public health centre clinics were invited to participate	Non-randomised controlled trial	Active: 49 Control: 54	**Duration: 6 weeks** **Nurse-led multidisciplinary PR program** in 9 primary healthcare centres: 6 weeks, 2 h sessions/week (1 h theory, 1 h exercise). Nurses covered disease management education, emphasising adequate nutrition. A social worker and physiotherapist each conducted one session on anxiety and stress management and physical activity education. Patients received individual home exercise prescriptions.	Medical management, no other intervention
**Tania 2017 [89]**	Switzerland, Valais	Single group pre–post study	57	**Duration: 12 months** Intervention based on the Chronic Care Model and the Canadian programme “Living Well with COPD: A Plan of Action for Life” included: - Patient and self-management education: 6 weekly group sessions (90–120 min each) covering disease education, medication, breathlessness and stress management, exacerbation prevention/management, and lifestyle behaviours including physical activity, smoking cessation, healthy diet, and good sleep habits. Classes led by respiratory physiotherapist/nurse specialised in self-management support, with a pulmonologist/pharmacist for specific sessions. - Scheduled follow-ups (every 4–6 weeks). - Information to and training of healthcare professionals.	
**von Leupoldt 2008 [82]**	Germany, Hamburg	Single group pre–post study	210	**Duration**: **3 weeks** Multidisciplinary outpatient **PR** 6 h/day, 5 days/week consisting of: - Exercise (20 h, endurance training on a stationary cycle ergometer, treadmill, and arm cycle ergometer; strength training for upper, lower, trunk, and respiratory muscles). - Patient education (11 h). - Nutrition counselling (5 h). - Breathing therapy (10 h). - Relaxation therapy (5 h). - Psychosocial education (3 h). - Smoking cessation support (4 h).	
**Yohannes 2021 [94]**	Not specified	Single group pre–post study	165	**Duration: 8 weeks** **PR** comprising biweekly, 2 h sessions: weekly 1 h circuit training (strengthening and endurance aerobic exercises) and 1 h group seminar on education topics like nutrition, smoking cessation, chronic disease coping, anxiety, panic management, and relaxation.	
**Santana 2010** [83]	Brazil, not further specified	Single group pre–post study	Total: 41 Analysis split into 2 groups: (1) ex-smokers: 18 (2) current smokers: 23	**Duration: 3 months** **PR programme** (3 times/week, 60 min each, 36 sessions total) including physical training and monthly educational lectures covering disease aspects, daily living activities, energy conservation, body awareness, and nutrition. Smoking’s harmful effects and its role in symptom maintenance were discussed, but no standardised smoking withdrawal programme or adjuvant drug treatment was administered.	
**van Boven 2016 [85]**	The Netherlands (recruitment across the whole country)	Single group pre–post study	88	**Duration: 1 year** **The Medication Monitoring and Optimisation (MeMO) COPD intervention** involved pharmacies collaborating with local primary care teams (GPs, physiotherapists, dietitians, and primary care nurses). Pharmacists received training on COPD pharmacotherapy optimisation, guidelines, and referral criteria. The intervention included a patient counselling session followed by a second consultation at 3 months and active adherence monitoring at 6 and 9 months. Initial counselling covered inhalation instructions, medication use, adherence, smoking cessation, and self-management recommendations. Pharmacists were encouraged to recommend physical activity, with or without referral to a physiotherapist, and refer patients with a low (<21) or high (>30) BMI to a dietician.	
**Lewis 2019 [90]** **(pilot study)**	UK, Islington	Single group pre–post study	42	**Duration: 4 weeks** The intervention had 4 weekly 2 h sessions, supervised by a senior physiotherapist and rehabilitation assistant. Allied health professionals, nursing, and medical colleagues contributed to the education component. Each session consisted of: - Education: smoking cessation and signposting other services (healthy eating, smoking cessation). - Exercise: warm up, strength and endurance exercises, and cool down (≥45 min). - Closing and planning period: setting goals and action plans for the next week.	
**Clarke 2016 [87]**	South Africa, Worcester	Single group pre–post study	12 (out of 12, 5 dropped out)	**Duration**: **12 weeks** Weekly home visits by ≥1 members of an intervention team (home-based caregiver, medical student, physiotherapy or human nutrition student): - Medical student: psychoeducation on clinical COPD implications, exacerbation warning signs, self-care (correct inhaler use, lifestyle modification including smoking cessation). - Physiotherapy students: tailored exercise programme for lower/upper body strength and endurance. Supervised patients every second week, alternating with home-based caregivers. - Human nutrition students: provided weekly nutritional information materials for discussion. Evaluated biometrics for supplementation.	
**Boueri 2001 [79]**	USA, Colorado, Denver	Single group pre–post study	37	**Duration: 3 weeks** **PR** included 12 exercise sessions with bicycle ergometer, upper-extremity and strength training, and stretching. Classes and reading material enhanced problem solving. Subjects were encouraged to attend nine group classes: COPD understanding, self-management, nutrition, stress management, breathing techniques, exercise importance, respiratory medications, oxygen therapy, and sexuality. Social worker-led individual/group sessions, with the patient and family, addressed psychological aspects like fear of death, guilt, depression, anxiety, and family relationships.	
**Sahin 2016 [88]**	Turkey, not further specified	Single group pre–post study Results reported separately for patients with low exacerbation risk (group 1) and those experiencing frequent exacerbations (group 2)	Total: 82 Group 1: 52 Group 2: 30	**Duration: 8 weeks** Outpatient **PR** programme 2 times/week consisting of supervised exercise training, theoretical training, nutritional intervention, and psychological counselling if needed. Exercises included breathing exercises, treadmill (≥15 min) and cycle training (≥15 min), peripheral muscle training, and stretching exercises. Patients also received home exercises.	
**Helvaci 2019 [91]**	Turkey, Ankara	Single group pre–post study	30	**Duration: 8 weeks** **COPD education and counselling programme (COPD-ECP)** based on Turkish Thorax Society recommendations and existing literature, provided an education booklet emphasising: a healthy lifestyle including balanced nutrition, adequate sleep-rest, harmful effects of smoking, benefits of smoking cessation, and various ways to cope with stress. The intervention, delivered by a registered nurse, included home visits for the first 4 weeks explaining booklet chapters (1 h) followed by discussion and clearing doubts (15 min). Telephone follow-ups addressed patient queries for the next 4 weeks.	
**Gagné 2020 [92]**	Canada, Québec	Single group pre–post study	54	**Duration: not specified** Respiratory educators engaged in a 7 h lecture-based continuing education (CE) activity on self-management support (SMS). Four months later, educators provided SMS to individuals with COPD in their everyday practice, incorporating components based on the PRISMS taxonomy: 1. Training for practical self-management activities. 2. Providing action plans for COPD exacerbation management. 3. Offering advice and support around lifestyle, including smoking cessation, nutrition counselling, stress and anxiety management, breathing techniques, and energy conservation. 4. Regular clinical reviews, e.g., to perform follow-up visits.	
**Benzo 2013 [84]** **(pilot study)**	USA, Minnesota	Single group pre–post study (pilot study)	11	**Duration: 8 weeks** Weekly in-person sessions featuring self-management coaching (involving motivational interviewing) for 60 min in the first session and 30 min subsequently, along with 60 min of exercise training. Implemented by 2 interventionists (1 registered nurse and 1 respiratory therapist). - First session focused on key COPD management behaviours, including incorporating daily physical exercises (3 or 4 upper extremity and lower extremity movements). - Following 7 sessions: self-management action planning in which patients select a self-management domain, including coping with fear and other emotions, quitting smoking, effective breathing, managing fatigue, coping with stress, taking medication, increasing physical activity, relaxation and positive thinking, relationships, and communication. Action planning continued by collaboratively setting a realistic goal that was completed in the next several days.	
**Kaplan 2004 [80]**	USA, not further specified (17 centres)	Single group pre–post study	1218	**Duration**: **6–10 weeks** **PR programme** with 16–20 supervised sessions at certified rehabilitation centres near participants’ homes. Components: (1) comprehensive evaluation of medical, psychosocial, and nutritional needs; (2) setting of goals for education and exercise training; (3) exercise training (i.e., lower extremity, flexibility, strengthening, and upper extremity); (4) education on emphysema, medical treatments; (5) psychosocial counselling; and (6) nutritional counselling.	
**Ngaage 2004 [81]**	UK, East Yorkshire	Single group pre–post study	14	**Duration: 6 weeks** Comprehensive **PR** program included: - Education and dietary advice: (1) basic lessons on the need for exercise, (2) dietary assessment and recommendations for caloric, fluid, and supplement intake, (3) disease education and dietary advice session. - Exercise programme: supervised by a physiotherapist in groups of 3–4 patients twice weekly. Circuit of 10 exercises targeting upper- and lower-body muscle groups, postural exercises, and general aerobic exercises. Home exercise programme included tolerated hospital exercises, performed 2–3 times daily on non-hospital days.	
**McDonald 2016 [86] & McLoughlin 2017 [96] (secondary analysis of McDonald et al., 2016 [86])**	Australia, Newcastle	Single group pre–post study	28	**Duration**: **3 months** **Combined diet and exercise intervention** included: - Diet: hypocaloric prescription (3850–5000 kJ/day, or up to 5900 kJ/day for BMI > 40 kg/m^2^). Two daily meal replacement supplements for BMI ≤ 40 kg/m^2^, three for BMI > 40 kg/m^2^, providing 870 kJ/serve (45% carbohydrate, 40% protein, 20% fat). Patients had a third balanced meal (1200–1750 kJ) and a further 900–1200 kJ daily to meet daily requirements. Recommended protein: 1.2–1.5 g/kg/adjusted body weight. Patients received ongoing dietitian counselling. - Exercise: Home-based strength training sessions, 3 days a week with a rest day, prescribed and supervised by a physiotherapist. Education and counselling optimised the intervention, with patients advised to maintain an exercise log.	
**Korkmaz 2020 [93]**	Turkey, Konya	Single group pre–post study	66	**Duration: 8 weeks** Two hour **PR** sessions, 3 days/week, included exercise and nutritional support: - 100–120 min of personalised, intense, and comprehensive endurance/strength exercises. - An expert dietician emphasised nutritional support, conducting body composition analysis to prevent weight loss and restore muscle atrophy. A personalised diet was determined. Patients with FFMI < 16 kg/m^2^ for men and <15 kg/m^2^ for women received oral nutritional therapy (125 mL liquid package) 3 times/day, with each serving containing 300 kcal, 18 g protein (E24%), 11.75 g fat (E35%), and 30.5 g carbohydrate (E41%).	
**Pagano 2023 [95]**	Australia, Syndey	Single group pre–post study	31	**Duration: 3 months** Physiotherapists completed an advanced training workshop in the management of COPD and coordinated a brief intervention in collaboration with general practice staff at three time points: - Referral to PR following COPD-X guidelines (national guidelines). - Physical activity advice and counselling using the 5 A’s approach (Ask, Advise, Assess, Assist, Arrange follow-up) according to the Australian Physical Activity and Sedentary Behaviour Guidelines. - Providing a pedometer and diary for monitoring PA goals. - Individualised smoking cessation advice and GP referral if needed. - Provision of education booklets on PA guidelines, smoking cessation, and COPD management. - Developing/reviewing COPD-specific GP management plans (GPMP) and/or action plans. Participants had follow-up visits at 1 month to review and set PA goals and a final assessment at 3 months to evaluate progress.	
**Ansari 2020 [34]**	Australia, Sydney	Single group pre–post study	50	**Duration: 6 months** **APCOM study**: self-management education programme for COPD in the context of multi-morbidity. PNs attended a 1-day workshop and were trained to deliver the self-management education programme: conduct a patient assessment to identify the patient’s health priorities using a template based on the Health Belief Model. The template included exercise recommendations, pulmonary rehabilitation, advice on overall health, smoking effects, benefits of quitting, and assistance with quitting. - First session: individual patient needs were assessed, and the intervention tailored accordingly. - Following 2 sessions: motivational interviewing addressed barriers in managing COPD with co-morbidities, working towards optimising health behaviours. Intervention delivery followed the 5As: Ask, Advise, Assess, Assist, and Arrange, according to the Australian Physical Activity and Sedentary Behaviour Guidelines. Health information and referrals were provided, as necessary. - After the last session, PNs followed up with patients via monthly phone calls for 5 months.	
**Asthma**					
**Vempati 2009 [97]**	India, Delhi	Randomised controlled trial	Active: 29 Control: 28	**Duration: 8 weeks** (2 weeks guided, then follow-up at home) **Yoga-based lifestyle modification and stress management programme:** Initial 2 weeks, 4 h sessions included practicing yoga and pranayama for 1 h supervised by a qualified yoga instructor, 30 min refreshment and group support building; 2-h lecture and discussion, and 30 min meditation. Education sessions included stress management and nutrition and health education by physicians. Participants received printed materials and audio cassettes to supplement live instruction. They were asked to maintain a daily diary on yoga practice, dietary advice, and rescue medication use, which was reviewed daily.	Received conventional care and were offered a session on health education relevant to their illness
**Ma 2015 [100]**	USA, Northern California	Randomised controlled trial	Active: 165 Control: 165	**Duration: 12 months** The **BEWELL intervention** aimed at modest weight loss and increased physical activity in three stages: intensive (13 weekly in-person group sessions over 4 months), transitional (2 monthly in-person individual sessions), and extended (≥3 bimonthly phone consultations based on participant needs). Theory-based and goal-oriented, staff counselled on healthy eating with moderate calorie reductions (500–1000 kcal/d, daily total not <1200 kcal), moderate-intensity activity (e.g., brisk walking), and behavioural self-management skills.	Usual care enhanced with a pedometer, a weight scale, and a list of routinely offered Kaiser Permanente in Northern California (KPNC) weight management services, and a KPNC standard asthma self-management educational DVD. The research team made no other attempts to intervene with control participants.
**Toennesen 2018** **[101]** **Bentzon 2019 [106] (Follow-up study of Toennesen et al., 2018 [101])**	Denmark, not further specified	Randomised controlled trial	Total: 125 Active: 29 Control: 34 There were 2 more intervention groups; therefore, active and control do not add up to total Total: 25 Active: 15 Control: 10	**Duration: 8 weeks** **Exercise + diet intervention**: - Exercise: high-intensity interval training on indoor spinning bikes 3 times/week, supervised by a qualified sports instructor. - Diet: 5 group counselling sessions (2–6 patients/group) and 1 individual counselling session with a trained study dietician. Diet advised: high protein (25–28% of energy), low glycaemic index, anti-inflammatory, i.e., higher amounts of vegetables, fruits, nuts, lean meat, fish, and seafood in accordance with an anti-inflammatory index.	Patients in the control group received no intervention and were encouraged to maintain usual physical activity levels and diet.
**Al-Sharif 2020 [102]**	Saudi Arabia, Jeddah	Randomised controlled trial	Active: 36 Control: 36	**Duration: 12 weeks** 45 min of treadmill-based aerobic exercise training (including 5 min warm up and 10 min cool down), 3 sessions/week for 12 weeks. Dietitian supervised diet regime providing 1200 Kcal/day.	No diet and exercise intervention
**Pokladnikova 2013 [99] (pilot study)**	The Czech Republic, not further specified	Randomised controlled trial	Active: 15 Control: 12	**Duration: 8 weeks** **Self-management programme**: 4 group meetings on yoga-based lifestyle changes (1.5 h each) and four individual sessions on psychotherapy based on Eastern philosophy and ethicotherapy (1 h each). Led by a certified yoga instructor and a psychotherapist specialised in spirituality-based cognitive-behavioural therapy. Lifestyle changes included an asthma-specific diet (Mediterranean style and allergen-free diet), yoga postures, stress management training encompassing relaxation, breathing techniques, meditation, emotion management, and communication skills training. Patients were told to practice lifestyle changes daily, receiving a workbook and diary for notes. Group sessions were educational, followed by discussion and skill training.	Standard care
**Tousman 2011 [98]**	USA, Virginia	Randomised controlled trial	Active: 21 Control: 24	**Duration: 7 weeks** **Asthma self-management programme:** 7 weekly 2 h meetings, consisted of interactive discussions, problem solving, social support, and a behaviour modification procedure. Meetings included a 60 min individual status report and a 60 min discussion topic. Participants received self-management behaviour homework to be practiced on a regular basis prior to the next session, covering asthma-specific goals and general lifestyle goals (e.g., 20 min of relaxation and exercise). Participants were asked to mark down points each day for these behaviours. Results were shared during the next week’s individual status report and the group provided feedback. Facilitated by a team including a psychologist, certified asthma educator (clinical nurse specialist), occupational therapist, and physician assistant.	Not specified
**Rasulnia 2017 [103]**	USA, Durham, North Carolina	Single group pre–post study	40	**Duration: 12 weeks** - Printed material/tools: healthy eating booklet, interactive goal and barrier contemplation booklet, diet and activity tracking booklets, pedometer, and a magnet reinforcing healthy eating principles. - Curriculum of easy-to-read, written materials included the topics of exercise, healthy living, eating right, and stopping smoking. - Weekly engagement with health advisor/coach using motivational interviews to support the received information.	
**Johnson 2022 [105]**	USA, Vermont and Arizona	Single group pre–post study	43	**Duration: 6 months** **Online weight loss intervention** included: - Individualised calorie and fat intake goals and increased physical activity through weekly real-time group meetings led by a dietitian. - Weekly lessons and individualised nutrition education emphasised consuming a diet high in fruits, vegetables, and whole grains and low in fat, sugar, salt, and alcohol. - Participants recorded daily food intake, activity, and weight in MyFitnessPal, monitored by the facilitator providing weekly feedback. - Incremental goals were set for achieving the recommended 200 min of moderate-intensity exercise weekly by week 9, with brisk walking recommended for its accessibility and safety.	
**Mammen 2022 [104]**	USA, New York	Single group pre–post study	30	**Duration: 6 months** Multi-component programme for remote primary care management of asthma: (1) smartphone asthma symptom monitoring, (2) smartphone-based telemedicine follow-up and self-management training (SMT) with a nurse via Zoom video-conferencing, and (3) guideline-based clinical decision support software for calculating asthma severity, control, and recommend step-wise therapy based on Expert Panel Report-3 guidelines. Telemedicine follow-ups and SMTs, led by a nurse, occurred every 2–6 weeks until asthma was well controlled and follow-up occurred every 2–3 months after asthma control was achieved. SMT included smoking cessation and exercise modules.	

BMI: body mass index, COPD: chronic obstructive pulmonary disease, FFMI: fat free mass index, GP: general practitioner, PNs: practice nurses, PR: pulmonary rehabilitation.

**Table 2 nutrients-16-01515-t002:** Reported lifestyle factor targets and outcomes of the described combined lifestyle interventions in the included studies. The colour green was used to signify a significant desired effect on the outcome and grey was used to signify no change in outcome (non-significant effects). Studies are reported based on the number of lifestyle factors targeted (high to low) and ordered according to study design.

Authors, Year, Reference	Intervention Targets						Moments in Time When Outcomes Were Measured after Baseline	Outcomes											
	Diet	Physical Activity	Smoking Behaviour	Alcohol Consumption	Stress Management	Sleeping Behaviour		Eating Behaviour	PA Level	Smoking Behaviour	Alcohol Consumption	Stress Level	Sleeping Behaviour (Sleep Quality)	FEV1	Symptoms/Dyspnoea	BMI/Weight	Exercise Capacity	Muscle Strength	QoL
COPD																			
RCTs																			
Kheirabadi 2008 [56]	✓	✓	✓		✓	✓	8 weeks												
							3 months												
Walters 2013 [61]	✓	✓	✓	✓	✓		12 months												
Wilson 2015 [65]	✓	✓	✓		✓		12 months												
Jonsdottir 2015 [67]	✓	✓	✓			✓	12 months												
Finnerty 2001 [54]	✓	✓			✓	✓	12 weeks												
							24 weeks												
Bendstrup 1997 [52]	✓	✓	✓				12 weeks												
							24 weeks												
Dheda 2004 [55]	✓	✓	✓				6 months												
Theander 2009 [57]	✓	✓	✓				12 weeks												
van Wetering 2010 [58]	✓	✓	✓				4 months												
							24 months												
Zwar 2012 [59]	✓	✓	✓				12 months												
Kruis 2014 [107]	✓	✓	✓				24 months												
Zwar 2016 [35]	✓	✓	✓				12 months												
Zhang 2020 [73]	✓	✓	✓				3 months												
							6 months												
							12 months												
							24 months												
Zhu 2021 [74]		✓	✓	✓			6 months												
Mitchell 2014 [62]	✓	✓			✓		6 weeks												
							6 months												
Benzo 2016 [69]		✓	✓		✓		6 months												
							12 months												
Gurgun 2013 [60]	✓	✓					8 weeks												
Lou 2015 [66]		✓	✓				4 years												
Suhaj 2016 [68]		✓	✓				6 months												
							12 months												
							18 months												
							24 months												
Markun 2018 [33]		✓	✓				1 year												
Jolly 2018 [70]		✓	✓				6 months												
							12 months												
Thom 2018 [71]		✓	✓				9 months												
Aboumatar 2019 [72]		✓	✓				6 months												
Emery 1998 [53]		✓			✓		10 weeks												
Blumenthal 2014 [64]		✓			✓		16 weeks												
Bourne 2022 [75]	✓	✓			✓		6 months												
							9 months												
Non-randomised controlled trials																			
Monteagudo 2013 [76]	✓	✓	✓				12 months												
Da Silva 2018 [78]	✓	✓	✓				12 weeks												
Zakrisson 2011 [77]	✓	✓			✓		1 year												
Single group pre–post studies																			
Tania 2017 [89]	✓	✓	✓		✓	✓	12 months												
von Leupoldt 2008 [82]	✓	✓	✓		✓		3 weeks												
Yohannes 2021 [94]	✓	✓	✓		✓		8 weeks												
							2 years												
Santana 2010–Ex-smokers [83]	✓	✓	✓				3 months												
Santana 2010–Current smokers [83]																			
van Boven 2016 [85]	✓	✓	✓				12 months												
Lewis 2019 [90]	✓	✓	✓				4 weeks												
Clarke 2016 [87]	✓	✓	✓				12 weeks												
Boueri 2001 [79]	✓	✓			✓		3 weeks												
Sahin 2016 ^a^ [88]	✓	✓			✓		8 weeks												
Helvaci 2019 [91]	✓		✓			✓	8 weeks												
							12 weeks												
Gagné 2020 [92]	✓		✓		✓		6 months												
Benzo 2013 [84]		✓	✓		✓		8 weeks												
Kaplan 2004 [80]	✓	✓					end of PR ^b^												
Ngaage 2004 [81]	✓	✓					6 weeks												
McDonald 2016 [86] & McLoughlin 2017 [96]	✓	✓					3 months												
Korkmaz 2020 ^c^ [93]	✓	✓					8 weeks												
Pagano 2023 [95]		✓	✓				3 weeks												
Ansari 2020 [34]		✓	✓				6 months												
Asthma																			
RCTs																			
Vempati 2009 [97]	✓	✓			✓		8 weeks												
Ma 2015 [100]	✓	✓					12 months												
Toennesen 2018 [101]	✓	✓					8 weeks												
Bentzon 2019 [106]							1 year follow-up												
Al-Sharif 2020 [102]	✓	✓					12 weeks												
Pokladnikova 2013 [99]	✓				✓		2 months												
							6 months												
Tousman 2011 [98]		✓			✓		7 weeks												
Single group pre–post study																			
Rasulnia 2017 [103]	✓	✓	✓				12 weeks												
Johnson 2022 [105]	✓	✓		✓			6 months												
Mammen 2022 [104]		✓	✓				6 months												

^a^ Improvement reported post-intervention in frequent exacerbators and non-exacerbators; ^b^ Completion of 16 to 20 PR sessions (over 6 to 10 weeks) was considered as the end of rehabilitation; exact timepoint not mentioned; ^c^ Significant improvements in body composition, i.e., increased fat free mass and reduced fat mass; no change in body weight post intervention; BMI: body mass index, COPD: chronic obstructive pulmonary disease, FEV1: forced expiratory volume in 1 s.

## 4. Discussion

This systematic review provides a comprehensive overview of existing CLIs for the management of asthma and COPD. Quite some research has already been performed on CLIs for the management of these conditions, but no studies on CLIs were identified for the prevention of the onset of asthma or COPD. The combination of diet and PA was identified as the most frequent target in CLIs for both asthma and COPD management, and in the case of COPD, this was often combined with smoking cessation. QoL was the most frequently reported outcome in CLIs for COPD management, and in CLIs for asthma management, both QoL and lung function (FEV1) were most frequently reported. The existing CLIs were shown to have beneficial effects on QoL and physiological outcomes, including respiratory symptoms, BMI/weight, and in the case of COPD, also exercise capacity. Lung function was shown to be retained after participation in CLIs. The influence of CLIs in terms of behavioural change was less well established because many studies did not report behavioural outcomes. If they did, improvements were mainly seen in dietary intake and PA level. None of the studies reported an undesired effect on any of the outcomes.

Besides the combination of diet, PA, and in the case of COPD, also smoking cessation, being targeted most frequently, there was less attention paid to other lifestyle factors (i.e., alcohol consumption, stress level, and sleeping behaviour) in the studied CLIs. There was less evidence for the relationship between these lifestyle behaviours and asthma or COPD development/progression; however, they are receiving increasing attention in public health, particularly in the prevention of other diseases [108,109,110]. Furthermore, studies that targeted alcohol consumption, stress management, or sleeping behaviour in this review often found improvements in QoL. It is therefore important that studies also include these lifestyle factors in their CLIs to make sure health is targeted in the most comprehensive way.

Pulmonary rehabilitation programmes showed the most promising results in COPD management. PR programmes are effective; however, these effects are often not maintained in the long term [111]. The PR programmes included in this review had a maximum duration of 3 months. It is therefore questionable whether the included PR programmes would have led to actual sustained behavioural changes. In general, it would be expected that in longer lasting interventions, in which individuals continue performing the behaviour for a longer amount of time, there would be a smaller chance of relapse. In this regard, it is also important to check whether behavioural change is maintained once participants are let loose of any type of intervention contact. Consequently, research in recent years has focused increasingly on maintenance PR, which is implemented after an initial PR programme [111], but the current review showed that the effects are still inconsistent [58,65]. This review shows the limited availability of long-term interventions as well as studies measuring follow-up outcomes, especially related to behavioural outcomes, after interventions end. During the search, even more articles on PR were obtained; however, many were excluded since exercise training was often the only targeted component. Based on the definition of PR, these programmes could potentially be considered CLIs. However, in reality, the focus of these programmes is often only on exercise training [26]. For asthma management, the most effective studies usually included dietary interventions, often also combined with a PA component. As study participants in all of these studies, except one, were overweight or obese, the observed effects on physiological outcomes and QoL could be attributed to improvements in weight, but weight was not measured in all of these studies.

Our literature search demonstrated that CLIs evaluating the effects on physiological outcomes clearly outran those reporting behavioural outcomes. Furthermore, those that did report effects on behavioural outcomes often did not seem to find effects. Smoking cessation was one of the most frequently targeted lifestyle factors, in combination with diet and PA for COPD management, but available CLIs may not be sufficient to induce a change in smoking behaviour; 90% of the included CLIs were ineffective in this regard. While smoking cessation is one of the most effective ways to reduce COPD morbidity, successful quitting in patients with COPD is difficult [112]. The percentage of current smokers among asthma patients was considerably lower compared to patients with COPD, which may reflect why smoking cessation was only targeted in one study for asthma management. A recent umbrella review, however, highlighted smoking as a risk factor for adult-onset asthma [13].

The effects of CLIs on physiological outcomes were more robust. More than half of the studies reporting on FEV1 levels in COPD found no effect, as expected, since it was unlikely that participation in a CLI would improve irreversible airflow obstruction. It was therefore unexpected that some studies did find improvements in FEV1 level, and the reasons for this are unknown. Stable FEV1 levels can also be considered desired, as FEV1 levels in patients with COPD usually worsen over time [113]. Throughout the duration of most CLIs, major decreases in FEV1 level were, however, not expected [114]. The majority of studies in both asthma and COPD management found improvements in BMI/weight. One study that did not find an effect on BMI, did find positive effects on fat free mass and fat mass change. Future studies should therefore consider reporting on different body composition parameters, besides BMI, as fat-free mass and fat mass are partly independently related to extra-pulmonary features. Exercise capacity was one of the most frequently reported outcomes for COPD, whereas it was not prioritised in the studies for asthma management. A plausible explanation for this could be that a decrease in exercise capacity is less prominent in patients with asthma compared to COPD [115]. The implementation of CLIs was shown to improve QoL in patients; both for asthma and COPD, the majority of studies reported a desired effect on QoL.

Overall, the methodological quality of the asthma studies was low, with the majority of studies rated as weak and only two studies rated as strong, even though 67% of studies were RCTs. Taken together with the lower number of CLIs available for asthma management, more studies are needed in order to draw more firm conclusions about their effects. For COPD, the majority of studies were also RCTs, but again, the methodological quality was independent of study design. More than half of the studies with a pre–post design were rated as strong, but no causal effects could be drawn from these studies. Even though many CLIs for COPD management have been studied, RCTs of higher methodological quality are needed, specifically paying attention to the prevention of selection bias and confounding.

The number of CLIs available for COPD management was considerably higher than that for asthma management. CLIs for asthma management increased in number mostly from the beginning of the last decade, with the first paper being published in 2009, while for COPD, it was published in 1997. Twenty-five percent of the studies were performed in Asia, South America, and Africa. Studies in low- and middle-income countries are important, as mortality and morbidity of asthma and COPD in these countries are highest [116,117].

An extensive search of the existing literature yielded no studies describing CLIs for the prevention of the onset of asthma or COPD. The primary focus of the CLIs was on secondary and tertiary prevention, even though the existing literature suggests an optimal lifestyle not only slows lung function decline in patients but can also play a role in reducing asthma and COPD risk [13,14,15,16,17,18,19,20,21,22,23]. As this review showed that CLIs are effective in asthma and COPD management, one could question whether the implementation of CLIs would not already be beneficial at an earlier stage. Perhaps a lack of awareness needs to be addressed first, both for high-risk groups being unaware of their risks and intermediaries being unaware of the possible benefits CLIs can have in this target group. A great deal may be gained by implementing CLIs in high-risk groups for the prevention of onset of asthma and COPD.

This comprehensive overview of existing CLIs for asthma and COPD management provides insight into what types of interventions already exist and which outcomes they affect positively (QoL, respiratory symptoms, BMI, exercise capacity (in COPD), and possibly lung function). This is helpful for the development and implementation of new interventions and helps to identify relevant intervention targets. Furthermore, intervention targets for which few data are available, including alcohol consumption, stress management, and sleeping behaviour, as well as the effects on outcomes about which little is known, especially behavioural outcomes, warrant further investigation.

The extensive scope of this review led to many articles and the wide variation among the selected studies, including heterogeneity in intervention types, implementation methods, and outcome variables, did not support the possibility of conducting a meta-analysis. The aim of this review was, however, to obtain an overview of which CLIs exist and conducting a meta-analysis would require comparable studies or groups. Furthermore, due to the wide scope, outcomes related to healthcare use (e.g., exacerbations, hospitalisations, drug use) were not included.

## 5. Conclusions

A substantial amount of CLIs are available that target behavioural changes in at least two lifestyle factors for the management of COPD, but fewer so for asthma management.

Despite the methodological limitations of individual studies, the effects of these CLIs on QoL and physiological outcomes were robust, leading to improvements in QoL, respiratory symptoms, BMI/weight, exercise capacity (COPD), and retained lung function. It is therefore recommended that more attention is given to the implementation of CLIs in clinical practice. However, no firm conclusions can be drawn about the effects on behavioural changes. Future studies should consider aiming for and reporting the effects on long-term behavioural lifestyle changes. Furthermore, CLIs should be implemented before disease onset in high-risk groups to study their effectiveness in the prevention of asthma and COPD.

## Figures and Tables

**Figure 1 nutrients-16-01515-f001:**
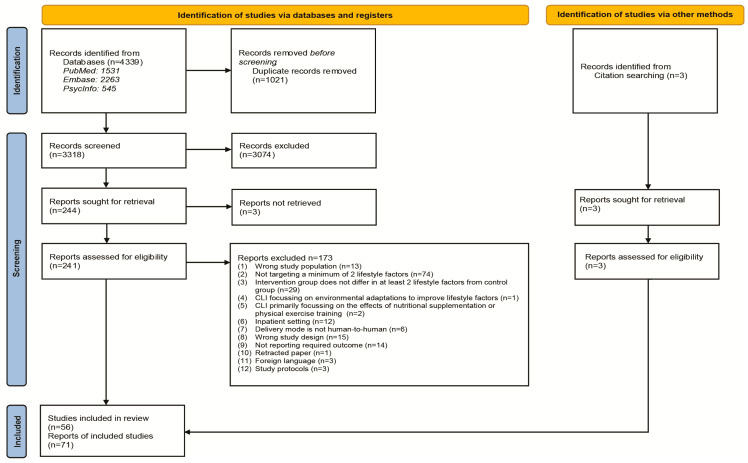
Prisma flowchart: selection of included studies. CLI: combined lifestyle intervention.

**Figure 2 nutrients-16-01515-f002:**
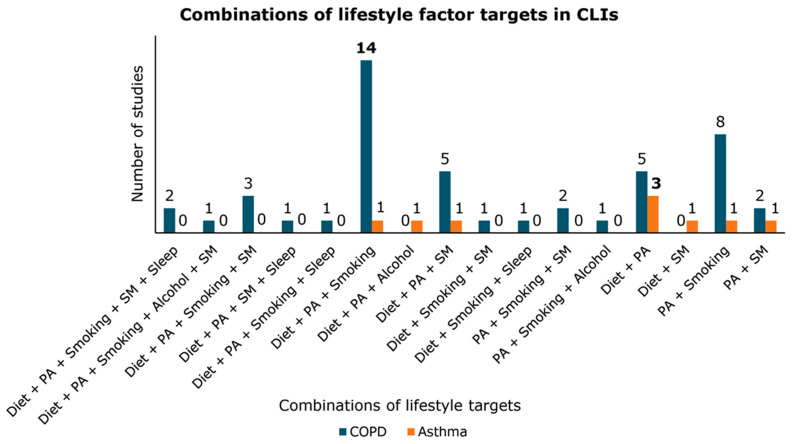
Combinations of lifestyle factor targets in CLIs. CLI: combined lifestyle intervention, COPD: chronic obstructive pulmonary disease, PA: physical activity, SM: stress management.

**Figure 3 nutrients-16-01515-f003:**
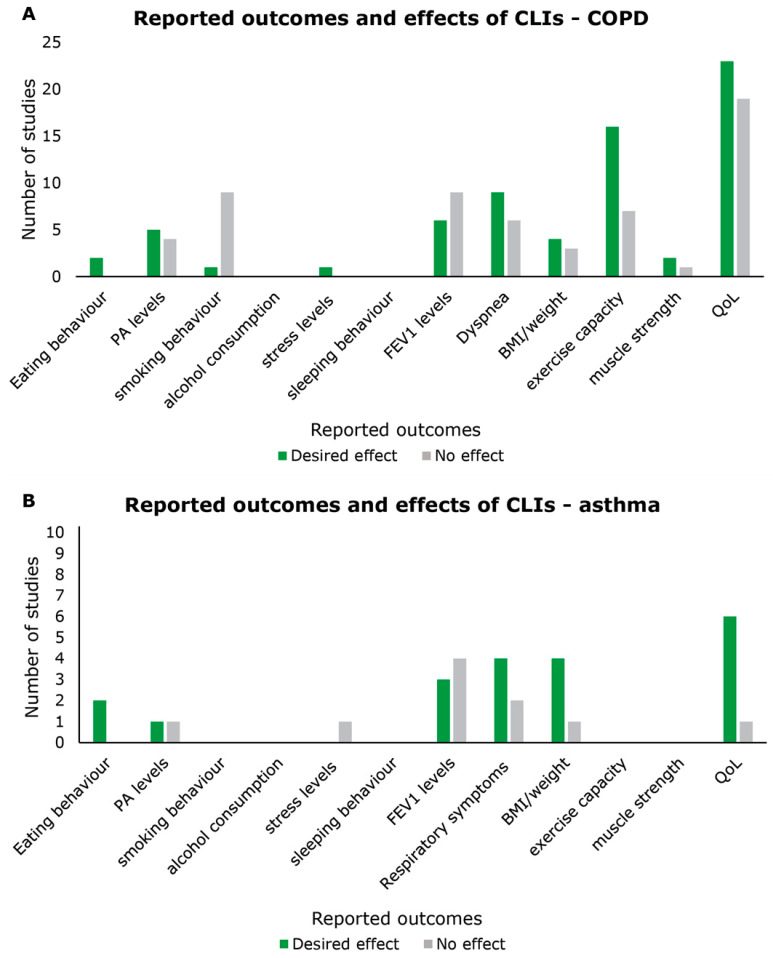
(**A**) Reported outcomes and effects of CLIs for COPD management; (**B**) Reported outcomes and effects of CLIs for asthma management; BMI: body mass index, CLI: combined lifestyle intervention, FEV1: forced expiratory volume in 1 s, PA: physical activity, QoL: quality of life.

## Data Availability

All data relevant for this study are available in this manuscript or the Appendix A. The data extraction form and quality assessment table are available on https://osf.io/2quxa/.

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
