# Peer review of "Combined Lifestyle Interventions in the Prevention and Management of Asthma and COPD: A Systematic Review"

_nutrients, 2024, doi:10.3390/nu16101515_

Round 1
Reviewer 1 Report
Comments and Suggestions for Authors
The paper titled "Combined lifestyle interventions in the prevention and management of asthma and COPD: A Systematic Review" presents a comprehensive analysis of various combined lifestyle interventions (CLIs) and their effects on asthma and chronic obstructive pulmonary disease (COPD). The authors have provided good background information on the relevance of lifestyle changes in managing COPD and asthma. Overall, the review is well-structured, adhering to PRISMA guidelines, which enhances its credibility. However, the author should address the following issues, which could help improve the significance of the study:
1. Kindly remove “on 24 August 2023” from the abstract; it seems to be redundant as in the methodology section, it has been mentioned as “performed from 22 June 2022 until 24 August 2023.”
2. The search strategy could be detailed more explicitly by specifying the custom range of the search. Was it from the date of inception, or did you follow a specific range? For example, from 1990 to 24 August 2023.
3. Rearranging the discussion will do more justice to the manuscript if you could bring the key findings at the beginning and end with the limitation.
4. While the conclusions mention the importance of lifestyle interventions, they fall short of providing actionable recommendations for practitioners. The paper could be strengthened by offering specific guidelines or protocols that could be implemented in clinical practice based on the evidence reviewed.
Author Response
We thank the reviewer for the thoughtful evaluation of our manuscript and for acknowledging the comprehensiveness of our analysis. We are pleased to learn that our manuscript is well-structured and adhered effectively to the PRISMA guidelines, thereby enhancing its credibility. Below we provide a point-by-point reply to the reviewer’s comments.
The paper titled "Combined lifestyle interventions in the prevention and management of asthma and COPD: A Systematic Review" presents a comprehensive analysis of various combined lifestyle interventions (CLIs) and their effects on asthma and chronic obstructive pulmonary disease (COPD). The authors have provided good background information on the relevance of lifestyle changes in managing COPD and asthma. Overall, the review is well-structured, adhering to PRISMA guidelines, which enhances its credibility. However, the author should address the following issues, which could help improve the significance of the study:
- Kindly remove “on 24 August 2023” from the abstract; it seems to be redundant as in the methodology section, it has been mentioned as “performed from 22 June 2022 until 24 August 2023.”
We thank the reviewer for this suggestion, this part has been removed from the abstract (lines 21-22). - The search strategy could be detailed more explicitly by specifying the custom range of the search. Was it from the date of inception, or did you follow a specific range? For example, from 1990 to 24 August 2023.
We agree with this comment. We therefore specified the dates of the search and added that no publication date limit was implemented in the search (lines 95-98): The initial search was performed on 22 June 2022 and repeated on 24 August 2023 to check whether new articles had been published in the past year. In order to get a complete overview of the available literature, no publication date limit was implemented. - Rearranging the discussion will do more justice to the manuscript if you could bring the key findings at the beginning and end with the limitation.
By mistake a sentence from the Word template provided by Nutrients was left in the submitted manuscript. This was the first sentence of the discussion, by removing this sentence, the discussion now starts with the key findings (lines 349-362). Furthermore, we have switched around the order of the last two paragraphs so the discussion now ends with the limitations of the review (lines 449-463). - While the conclusions mention the importance of lifestyle interventions, they fall short of providing actionable recommendations for practitioners. The paper could be strengthened by offering specific guidelines or protocols that could be implemented in clinical practice based on the evidence reviewed.
Due to large heterogeneity in the CLIs studied (e.g. in duration of the interventions or the (combinations of) lifestyle factors targeted) it is difficult to draw very specific recommendations for practitioners. We do understand the point raised by the reviewer and therefore we have added a more general recommendation to the conclusion. This to indicate that more attention should be given to the implementation of CLIs in clinical practice, as they have shown to be effective, which in our view is the most important next step needed in practice (lines 469-471): It is therefore recommended that more attention is given to the implementation of CLIs in clinical practice.
Reviewer 2 Report
Comments and Suggestions for Authors
I read it with great interest, but I have raised several concerns.
#1. Chronic respiratory diseases (CRDs) are among the most common non-communicable diseases worldwide -> Please cite the land-mark study.
#2. In Ref 29, please cite the appropriate paper related on PRISMA 2020 guideline.
#3. effect can only use the RCT results. Please use the association.
#4. Please add the graphical figure to improve the readability.
#5. Please describe the searching terminology.
#6. Considering that the evidence of most papers is not high, I think the overall tone down of the papers is necessary.
Comments on the Quality of English LanguagePlease receive the English editing service.
Author Response
We thank the reviewer for the thoughtful evaluation of our manuscript and feedback on our work. Below we provide a point-by-point reply to the reviewer’s comments.
I read it with great interest, but I have raised several concerns.
- Chronic respiratory diseases (CRDs) are among the most common non-communicable diseases worldwide -> Please cite the land-mark study.
We thank the reviewer for this suggestion, the reference has been adjusted to the Global status report on noncommunicable diseases 2010 (lines 506-507): World Health Organization. Global status report on noncommunicable diseases 2010. Geneva, Switzerland; 2011. Accessed 7 May, 2023. This was the reference that was also cited to in the previous used reference (Global action plan for the prevention and control of noncommunicable diseases 2013-2020). - In Ref 29, please cite the appropriate paper related on PRISMA 2020 guideline.
We thank the reviewer for this thorough comment. An error had occurred with the numbering in the reference list, after adding an enter line, the reference list has now been adjusted and the numbers match the correct reference. In Ref 29, the appropriate paper related to the PRISMA 2020 guidelines is now cited (lines 568-569). - Effect can only use the RCT results. Please use the association.
We agree that only RCTs can conclude on effects, however, for clarity we described all findings of the included studies in the results section without distinguishing in methodological quality. We have also addressed this in the paragraph on methodological quality in the discussion (lines 422-431). - Please add the graphical figure to improve the readability.
This comment is unclear to us, we are not sure what the reviewer refers to here. If the reviewer is referring to Figure A3 in the appendix, we are happy to add this figure to the manuscript itself. If the reviewer is referring to something else, we ask if they could clarify. - Please describe the searching terminology.
The search terms are briefly described in lines 98-99 of the methods section and are presented in more detail in the Appendix (including the complete search strategies). - Considering that the evidence of most papers is not high, I think the overall tone down of the papers is necessary.
Based on this comment, we have adjusted the conclusion to some degree (line 467): Despite methodological limitations of individual studies, the effects of these CLIs on QoL and physiological outcomes were robust; leading to improvements in QoL, respiratory symptoms, BMI/weight and exercise capacity (COPD), and retained lung function. Furthermore, we believe lines 471-472 also tone down the results of the papers.